# Morpho-Physiological and Transcriptome Changes in Tomato Anthers of Different Developmental Stages under Drought Stress

**DOI:** 10.3390/cells10071809

**Published:** 2021-07-17

**Authors:** Anthony Tumbeh Lamin-Samu, Mohamed Farghal, Muhammad Ali, Gang Lu

**Affiliations:** 1Department of Horticulture, College of Agriculture and Biotechnology, Zhejiang University, Hangzhou 310058, China; anthonylaminsamu@yahoo.com (A.T.L.-S.); mohamedfa2016@gmail.com (M.F.); 2Department of Biological Sciences, Faculty of Pure and Applied Sciences, Fourah Bay College, University of Sierra Leone, Mount Aureol, Freetown 232, Sierra Leone; 3Key Laboratory of Horticultural Plant Growth, Development and Quality Improvement, Ministry of Agricultural, Zhejiang University, Hangzhou 310058, China

**Keywords:** abscisic acid, abiotic stress, development, drought, phytohormones, pollen, tomato, transcriptome

## Abstract

Drought limits the growth and productivity of plants. Reproductive development is sensitive to drought but the underlying physiological and molecular mechanisms remain unclear in tomatoes. Here, we investigated the effect of drought on tomato floral development using morpho-physiological and transcriptome analyses. Drought-induced male sterility through abnormal anther development includes pollen abortion, inadequate pollen starch accumulation and anther indehiscence which caused floral bud and opened flower abortions and reduced fruit set/yield. Under drought stress (DS), pollen mother cell to meiotic (PMC-MEI) anthers survived whereas tetrad to vacuolated uninucleate microspore (TED-VUM) anthers aborted. PMC-MEI anthers had lower ABA increase, reduced IAA and elevated sugar contents under DS relative to well-watered tomato plants. However, TED-VUM anthers had higher ABA increase and IAA levels, and lower accumulation of soluble sugars, indicating abnormal carbohydrate and hormone metabolisms when exposed to drought-stress conditions. Moreover, RNA-Seq analysis identified altogether >15,000 differentially expressed genes that were assigned to multiple pathways, suggesting that tomato anthers utilize complicated mechanisms to cope with drought. In particular, we found that tapetum development and ABA homeostasis genes were drought-induced while sugar utilization and IAA metabolic genes were drought-repressed in PMC-MEI anthers. Our results suggest an important role of phytohormones metabolisms in anther development under DS and provide novel insight into the molecular mechanism underlying drought resistance in tomatoes.

## 1. Introduction

Water deficit is one of the most important abiotic stresses restricting plant production. As a consequence of climate change, drought events are projected to increase in intensity, duration, and frequency [1]. Thus, the improvement of drought-resistance of crops represents an urgent need that demands the identification of key regulators and pathways as potential targets for drought-resistance improvement. Drought stress at the reproductive stage causes more severe damage and yield loss than at any other stage of development of crop plants [2,3]. Therefore, an effective way to improve the drought resistance of crops is to select for yield and its components during reproductive development under drought stress. Reduction in grain yield due to drought has mainly been attributed to male sterility because the male organ is more drought sensitive than the female organ which remains fertile under stress conditions that cause sterility in the male [4,5]. Hence, the male development under drought has attracted greater research attention than the female. Commercially, tomato is the most important vegetable crop. However, it is susceptible to abiotic stresses including drought. An earlier study examined the effects of drought during reproductive development in tomatoes utilizing vegetative tissues [6]. However, for a fruit vegetable crop like tomato, the fruit set is the most important trait for evaluating drought tolerance, thus evaluating drought tolerance of the reproductive organs especially the male is of enormous importance.

Anther, the male organ of flowering plants comprises concentric cell layers including the epidermis, endothecium, middle layers and tapetum, the innermost layer surrounding a central locule that contains sporogenous cells. Sporogenous cells develop into pollen grains, the male gametophytes, within the locule. During pollen development, the anther wall layers play important roles in nutrition, protection and pollen release. The tapetum serves as a source of energy for the developing microspores, secrete enzymes, e.g., callase that releases the tetrads from the callose wall [7] and precursors for exine wall formation. Initially, tapetum development proceeds normally and at a later stage undergoes programmed cell death (PCD) and disintegrates [8]. The timely tapetum-specific PCD and disintegration are necessary for normal pollen development. Precocious or postponed tapetum degeneration results in male sterility [9,10]. The modified epidermal cells, the stomia, modulate the pollen release process. Failure of stomia cells to degenerate leads to male sterility due to anther indehiscence [11]. Abnormalities in anther and pollen development leading to induction of male sterility as a result of drought were extensively investigated in cereal crops [12,13]. However, there is no information on the impacts of drought on anther and pollen development in tomatoes.

Carbohydrate metabolism and sugar movement from source organs to sink tissues such as anthers are important processes for pollen development. Studies with cereals have shown that disturbances in these processes result in male sterility associated with alteration in anther soluble sugar contents and, reduced/and or lack of starch accumulation in the pollen [14,15,16]. Sucrose accumulation, inadequate starch build-up in pollen grains and subsequent abortion of pollen development were attributed to repression of genes involved in sucrose and starch metabolism in drought- and cold-stressed anthers [15,16]. Additionally, drought-stressed rice anthers accumulate more starch granules in connective tissues than in pollen grains [17,18]. However, the underlying molecular mechanisms of sugar accumulation participating in drought stress responses during anther development remain unclear.

Phytohormones are important endogenous chemical messengers that modulate plant growth and development and respond to adverse stress factors. Among the phytohormones, abscisic acid (ABA) is the key plant hormone known to mediate responses to abiotic stresses such as drought and temperature [19]. In reproductive tissues, there is increasing evidence of a strong correlation between ABA level increase and pollen sterility during abiotic stresses. Higher pollen abortion and greater reduction in grain set occurred concomitantly with higher ABA accumulation in anthers of drought-susceptible than in drought-tolerant wheat cultivars [2,20]. Besides, there is proof of crosstalk between ABA and sugar signaling. The expression of *OSINV4*, a cell wall invertase gene, was repressed by ABA in cold-stressed rice anthers [21]. A recent study implicated auxin (IAA) in the control of plant response to abiotic stress during reproductive development. Reduction in pollen fertility and grain yield was associated with reduced accumulation of endogenous IAA in rice spikelets due to drought repression of YUCCA genes that are involved in IAA biosynthesis [22]. Jasmonic acid (JA) has been demonstrated to participate in drought stress responses in vegetative tissues [23,24] but there are no reports on JA mediation of drought responses in reproductive organs. However, induction of stigma exertion as a result of a reduction in endogenous JA in tomato anthers under high temperature was reported [25]. Although phytohormones play critical roles in regulating crops’ responses to drought stress, the dynamics of endogenous hormone metabolisms and the relationships with behavioral patterns of anthers and pollen at different stages of development under drought stress are not well understood.

In this study, we examined the effects of drought stress on anther morpho-physiological and molecular responses in tomatoes. The objectives of the study were to determine: (a) the consequences of subjecting anthers of varied developmental stages to water deficit on flowering, flower development, male gametophyte fertility, and fruit set/yield; (b) the histo-cytological changes in anthers and developing pollen at different stages of development under drought stress; (c) the effects of drought stress on changes in the levels of endogenous hormones and sugar accumulation in anthers at different developmental stages; and (d) the gene expression patterns in tomato anthers in response to drought stress at different developmental stages. Findings in this study contribute to our understanding of the associated physiological and molecular mechanisms underlying adaptation of anthers to drought stress in tomatoes.

## 2. Materials and Methods

### 2.1. Plant Material and Growth Conditions

In this study, the tomato (*Solanum lycopersicum*) cultivar ‘Micro-Tom’, provided by the Tomato Genetics Resource Centre (University of California, Davis, CA, USA), was used. Seeds were germinated on moist filter paper at 28 °C and planted in 72-well seed trays filled with soil that composed of peat, vermiculite, and perlite (4:2:1) and grown in a growth chamber with 25/20 °C (day/night) temperatures, constant 60% relative humidity, 16 h photoperiod and light intensity of 200 μmol m^−2^ s^−1^. After 3 weeks, uniform seedlings were planted singly in 0.8 L pots (height: 11 cm, diameter: 13 cm top, 8 cm bottom, without drainage holes) containing 150 g of the same soil and grown in the same growth chamber. Plants were grown under well-watered (WW) conditions until the reproductive stage and then divided into two treatment groups: WW, 65 ± 5% soil moisture; drought-stressed (DS), 6 ± 3% soil moisture). Each experiment was laid out in a completely randomized design with 3 replications, 12 plants for each treatment. At the binucleate stage of pollen development of the first floral bud (about 7 mm in length) (Appendix A), drought stress (DS) was imposed by withholding watering in the DS group. The soil moisture was allowed to decrease to 6% (Figure 1A) when signs of wilting were visible in the canopy leaves of DS plants. The soil moisture was maintained at 6 ± 3% for 4 days, and then normal rewatering resumed (Figure 1A).

Meanwhile, the well-watered (WW) plants were grown in the same chamber and watered (soil moisture of 65 ± 5%). Soil moisture was monitored using HH2 Soil Moisture Meter (Delta-T Devices, Cambridge, UK).

### 2.2. Phenotypic and Fruit Yield Characterization

For each plant, six inflorescences/trusses were tagged with different colored strings and the effect of drought stress was assessed for floral buds on tagged trusses only. The position of all floral buds was recorded and monitored for abortion, anthesis, opened flower abortion and fruit set. Flowering phenology was determined by counting the number of newly opened flowers per day from start to end of flowering. Floral bud abortion, fruit set and opened flower abortion were determined by expressing the total number of aborted floral buds, opened flowers that set fruit and opened flowers that aborted respectively as percentages of the total number of floral buds recorded on all tagged trusses. Fruit yield (g/plant) was determined at maturity. Each fruit was weighed using digital balance and only fruits with weights ≥ 1 g were considered. The means were estimated from 3 biological replications.

To determine whether DS affected the growth of floral buds, the lengths of floral buds at different stages of pollen development on tagged trusses were recorded at the time when soil moisture reached 6%. After the plants had grown at soil moisture 6 ± 3% for 4 days, the lengths of the same floral buds were determined again and recorded and the DS plants were rewatered. After rewatering, the length of their flowers was estimated at the date of anthesis.

### 2.3. Assessment of Gametophyte Fertility

To determine whether drought stress affected the fertility of the male gametophyte, the pollen viability of all flowers on tagged trusses in the WW and DS plants was determined, from start to end of flowering. Pollen viability was determined by staining of pollen with Alexander’s stain [26], observed, and photographed using Nikon Eclipse 90i microscope. Viable (red) and nonviable (white) pollen grains were counted from six different microscopic fields for each anther and used to calculate the percentage of pollen viability.

To determine whether DS affected the viability of female gametophyte, fruit set and flower abortion were further evaluated by performing artificial pollination experiments with the following treatment groups: (1) self-pollinated well-watered plants (WW **⊗**); (2) self-pollinated drought-stressed (DS) plants (DS **⊗**); (3) stigmas of flowers on WW plants cross-pollinated with pollen grains from DS plants (WW 
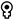
 × DS 
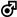
); and (4) stigmas of flowers on DS plants cross-pollinated with pollen grains from WW plants (DS 
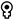
 × WW 
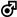
). The emasculation of female flowers was performed 1 d to anthesis and hand-pollinated the following evening using anthers from flowers that opened on the same day as a pollen source.

### 2.4. Histo-Cytological Studies

For histo-cytological observation, anthers at 7 different developmental stages: pollen mother cell (PMC), meiotic (MEI), tetrad (TED), early uninucleate microspore (EUM), vacuolated uninucleate microspore (VUM), binucleate (BIN) and mature pollen (MP) as previously described [27], were sampled from WW and DS plants immediately after 4 days of DS at 6 ± 3% (Figure 1C). The anthers were divided into three groups each containing anthers of different developmental stages based on similarity in response to the 4 days DS during phenotypic evaluation experiment: group 1 (PMC-MEI), contained the PMC and MEI anthers with length ≤ 3 mm; group 2 (TED-VUM), contained the TED, EUM and VUM anthers with lengths ≥ 4 mm to ≤ 6.4 mm; group 3 (BIN-MP), contained the BIN and MP anthers with length > 7 mm. The anthers were fixed in 2.5% glutaraldehyde and 1% Osmium tetroxide (OsO_4_) in phosphate buffer (PBS 0.1 M, pH 7.0). The samples were three times rinsed with phosphate buffer and dehydrated through an ethanol series from 30% to 100% and embedded in spur resin. Semi-thin sections were cut using LEICA EM UC7 utratome (LKB) and stained with 1% methylene blue and potassium iodide solution. The same samples were double-stained with uranyl acetate and alkaline lead citrate. Microscopic and transmission electron microscope observations were as described previously [27].

### 2.5. Transcriptome Analysis

#### 2.5.1. Samples for RNA-Sequencing

The anthers for transcriptomic analysis were sampled separately for groups 1, 2 and 3 from WW and DS plants after 4 days of DS at 6 ± 3% (Figure 1C) from 3 replicate experiments, immediately frozen in liquid nitrogen and stored at −75 °C until used.

#### 2.5.2. Preparation of RNA-Seq Library and Sequencing

Total RNA was extracted utilizing Trizol reagent (Invitrogen, Waltham, MA, USA). RNA quantity and quality were determined by NanoDrop 1000 spectrophotometer (Thermo Scientific Inc., Waltham, MA, USA), 1% agarose gel electrophoresis and Agilent 2100 Bioanalyzer (Agilent Technologies, Palo Alto, CA, USA). Following the protocol described by [28], strand-specific RNA-Seq libraries from 3 biological replicates for each group from WW and DS anthers were prepared using 1 ng/µL of total RNA sample and sequenced by Novogene Biotech (Beijing, China) on Illumina HiSeq 4000 system (Illumina, Inc., San Diego, CA, USA) according to the manufacturer’s instructions. The raw sequence reads were deposited into NCBI Sequence Read Archive under accession the number PRJNA746070.

#### 2.5.3. Analysis of RNA-Seq Data

Raw RNA-Seq reads were processed to remove sequences with adapter contamination, low-quality nucleotides in excess of 10%, and unknown nucleotides greater than 50 using Trimmomatic [29]. The remaining clean reads were aligned to tomato reference genome SL3.0 [30] using HISAT2 v2.0.5 [31]. The expression level of each gene was determined by counting the number of fragments that mapped to each gene and then normalized to the number of fragments per kilobase of transcript sequence per millions (FPKM) base pairs sequenced using HTSeq software. A gene with FPKM value ≥ 0.1 was considered expressed. Differentially expressed genes (DEGs) between WW and DS anthers were identified using DESeq software [32] with a rigid threshold of adjusted *p* < 0.05 and |log_2_ fold change| > 1. GOseq [33] was used to analyze the functional enrichment of specific gene ontology (GO) terms for DEGs. KEGG pathways significantly enriched with DEGs were determined using the KOBAS software [34].

### 2.6. Validation of RNA-Seq Data

Total RNA was extracted from samples using the plant RNA kit (OMEGA, Norcross, GA, USA) with three replications. cDNA was reverse-transcribed using PrimeScript™ RT Reagent Kit following the manufacturer’s protocol (Takara, Dalian, China) and then utilized for RT-qPCR reactions performed in Bio-Rad CFX96 (Bio-Rad, Berkeley, CA, USA) using SYBR^®^ Green Realtime PCR Master Mix. Gene-specific primers utilized in this study are listed in Appendix A. Relative gene expression levels were normalized utilizing the internal control gene *SlUbi3* and were calculated using 2^−ΔΔCt^ method [35].

### 2.7. Extraction and Quantification of Sucrose, Fructose and Glucose

Anthers were divided into three groups and sampled in the same manner as for the transcriptome analysis. Fresh anthers (200 mg) were ground into fine powder in liquid nitrogen and homogenized in 2 mL ethanol (80%). The samples were incubated at 80 °C for 30 min and centrifuged at 12,000 rpm for 20 min at room temperature. The precipitate was re-suspended in 80% ethanol and re-extracted. The supernatants were pooled and incubated at 90 °C until dryness, then 3 mL of ddH_2_O was added to the pellets and filtered using a 0.45 µm microporous membrane. High-performance liquid chromatography (HPLC) analysis of sucrose, glucose, and fructose contents was performed from three biological replicates of each sample as previously described [25].

### 2.8. Phytohormones Extraction and Quantification

Anthers were also divided into three groups and sampled in the same manner as for the transcriptome analysis. The extraction and purification of auxin (IAA), abscisic acid (ABA) and jasmonic acid (JA) were carried out following previously described methods [36,37] with slight modifications. Briefly, the sample was pulverized in liquid nitrogen. A total of 0.1g of powder was homogenized in 1 mL of ethyl-acetate containing 25 µL of internal standards of d2-IAA (Sigma-Aldrich, St. Louis, MO, USA), d5-JA (QCC) and d6-ABA (OlchemIm Ltd., Czechoslovakia), agitated for 12 h at 140 rpm at 4 °C, then centrifuged for 10 min at 12,000 rpm at the same temperature and the supernatant collected. The sample was re-extracted once with 1 mL ethyl-acetate. The supernatants were pooled, dried using nitrogen gas. The precipitate was re-suspended in 0.5 mL of 70% methanol and centrifuged for 10 min at 12,000 rpm at 4 °C. Three aliquots of 0.2 mL of sample supernatant were placed in separate snap-cap vials and analyzed using HPLC-mass spectrometry as previously described [25].

### 2.9. Data Analysis

Data were analyzed for statistically significant differences using Student’s *t*-test in Microsoft Excel 2019 or IBM SPSS Statistics 25 (IBM Corp., Armonk, NY, USA) using least significant difference at *p* < 0.05 level (LSD 0.05) and presented as means with standard deviations using GraphPad Prism, version 9.0 (GraphPad Software, Inc., CA, USA).

## 3. Results

### 3.1. Drought Stress Affects Flower Development and Fruit Yield

To assess the effect of drought on male reproductive organs, watering was suspended when the first emerged floral bud on the first truss was about 3 days to anthesis (Appendix A). The soil moisture was allowed to decrease to 6% after 6 days of watering suspension. At this time, designated as −4 days to rewatering (−4 DTRW), the DS period started (Figure 1A). The plants were grown at soil moisture 6 ± 3% for 4 days, then normal watering resumed at 0 days to rewatering (0 DTRW) (Figure 1A). At 0 DTRW, we observed that DS affected the growth and development of the plants and the developing anthers. The DS plants were shorter than the WW plants (Appendix A). Initially, flowering in both WW and DS plants was similar and reached its highest level at −3 days to rewatering (DTRW). Then the number of newly opened flowers/day in the DS plants was significantly reduced (5.2 and 1.3 /day compared with WW plants (14.0 and 15.3 /day) at −1 and 0 DTRW, respectively (Figure 1B).

Furthermore, at 0 DTRW, the −4 DAA floral bud reached anthesis whereas the −5 DAA floral buds remained unopened, though it was at an advanced stage in development, i.e., 1 DAA (Figure 1C, group 3 and Appendix A). One day after rewatering (1 DARW), the −1 DAA anther at 0 DTRW, reached anthesis (Figure 1B). From 2–7 DARW the production of newly opened flowers in the DS plants ceased completely while WW plants continued to produce newly opened flowers (Figure 1B). Surprisingly, the production of freshly opened flowers in DS plants began at 8/9 DARW (Figure 1D, group 1). However, we observed that in the DS plants, between the flowers that opened 1 and 8/9 DARW, much older floral buds in group 2 than those in group 1, at 0 DTRW (Figure 1C), were yet unopened at 8/9 DARW (Figure 1D) when flowering resumed in the younger floral buds and thus were declared aborted and exhibited irreversible growth inhibition (Appendix A). At 9–15 DARW, the number of freshly opened flowers in the DS plants was significantly higher than in the WW plants. During this period the production of newly opened flowers had ceased completely in the WW plants whereas the DS plants continued to produce new flowers albeit at lower rates until about 19 DARW (Figure 1B and Appendix A). This suggests that DS postponed flowering and caused bimodal flower production in DS plants.

To determine the effect of DS on the growth of the developing anthers, we measured the length of floral buds at different stages of pollen development at −4 and 0 DTRW, representing the start and end of the DS stress period, respectively (Figure 1A and Appendix A). We found that the lengths of the floral buds from DS plants were extremely shorter than those of the WW floral buds at 0 DTRW (Appendix A), indicating that DS impeded and reduced the growth and development of floral buds from the PMC-VUM stage. We also observed that the WW floral buds at 0 DTRW, were more advanced in development and significantly longer than their stages/lengths at −4 DTRW whereas the lengths of the DS floral buds at 0 DTRW were significantly shorter than their lengths at −4 DTRW (Appendix A). However, in the BIN-MP group, five floral buds that were −1 DAA (the most mature) to −5 DAA (the youngest) at −4 DTRW (Appendix A) continued to grow during the DS period with the −1, −2, and −3 DAA floral buds reaching anthesis at −3, −2, and −1 DTRW respectively. At 0 DTRW, the −4 DAA floral bud reached anthesis while the −5 DAA floral bud was still unopened but at an advanced stage of development i.e., −1 DAA (Figure 1C, group 3). This suggests that when floral buds are approaching the mature pollen/anthesis stage, DS doesn’t halt their growth.

The aborted floral buds were yellowish and distorted in appearance compared with the green and actively growing floral buds in the WW plants (Figure 1E). The specific developmental stages of the aborted anthers and those that resumed growth after rewatering were determined by assessing their positions and lengths at −4 DTRW, 0 DTRW and 8/9 DARW when flowering resumed (Appendix A). We found that the aborted buds were only among those recorded as TED (4 mm), EUM (5 mm) and VUM (6 mm) at −4 DTRW whereas those that resumed growth were among those recorded as MEI, PMC and younger buds (Figure 1C). These results suggest that drought stress at TED to VUM and MEI-PMC stages leads to irreversible and reversible growth inhibition and demonstrate the high sensitivity and resistance of these stages to DS respectively.

Among all the floral bud recorded in the DS plants, 25% aborted before anthesis whereas in the WW plants floral bud abortion was zero (Figure 1E). The aborted floral buds were distributed among three developmental stages only, as follows: TED, 40%; EUM, 42.5% and VUM, 7.5% (Figure 1E). Of the recorded floral buds in the DS plants that reached anthesis, 38.6% of the opened flower aborted thereafter which was markedly higher than 22.2% in the WW plants (Figure 1F). On the other hand, 36.4% of recorded floral buds in the DS plants reached anthesis and set fruit which was significantly lower than 77.7% in the WW plants (Figure 1F). Correspondingly, fruit yield/plant in the DS plants (40.5%) was noticeably lower than in WW plants (95.1 g/plant) (Figure 1F). Our data suggest that floral bud and opened flower abortions are major components of the contributing factors for the poor yield performance of the drought-stressed tomato plants. 

All opened flowers in the WW and DS plants were observed for morphological abnormalities. Surprisingly, we observed that among the flowers that began to open in the DS plants 8/9 DARW, were flowers with the stigmas exerted above the stamens (Appendix A). The stigma-exerted flower constituted 11.8% of all opened flowers in the DS plants, whereas none was found in the WW plants (Appendix A). To clarify whether the stigma exertion is due to elongation of the pistil or shortening of the stamen, we measured the lengths of the stamens and pistils in normal WW flowers and stigma-exerted flowers. We revealed that the stamen length in the exerted stigma flowers was significantly shorter (5.6 mm) than the pistil (7.2 mm) whereas the stamen length in the normal flower, though not significantly different, was longer (8.0 mm) than the pistil (7.8 mm) (Appendix A). This suggests that the exerted stigma phenotype was due to the shortening of the stamen and not elongation of the style. In short, drought stress affects tomato reproductive development in diverse ways.

### 3.2. Drought Stress Reduces the Fertility of the Male Gametophyte

Because drought stress decreased the overall fruit yield, we investigated whether abnormalities in the male organ development might play roles. We found that the average, pollen viability of all flowers in the DS plants was 66.6%, which was 31.9% lower than in the WW plants (97.8%) (Figure 2A) consistent with the overall reduced fruit yield. This suggests that abnormalities in the male organ development due to DS might have crucial roles in the reduction of fruit yield in the DS plants. To find out about the affected developmental stages that might have contributed to the overall low pollen viability, we examined the pollen viability of opened flowers daily as DS progressed from −4 to 0 DTRW. Surprisingly, we found that the pollen viability of flowers that opened in the DS plants from −4 to 0 DTRW was not significantly different from that in the WW plants (Figure 2B,C) suggesting that DS might not affect pollen viability in maturing anthers, −4 days to anthesis (DAA) and older (Appendix A).

In addition, we examined the pollen viability of flowers that opened after rewatering daily and we found that the flowers in the DS plants that opened one day after rewatering (1DARW) had pollen viability of 83.9% which was significantly lower than in the WW plants with a reduction of 14.8%. This suggests that among the floral buds that continued with growth during the drought stress, the transition to pollen mitosis 1 (TPM1) floral bud (−5 DAA) was more drought-sensitive (Appendix A). The floral buds that occurred from 2–7 DARW never reached anthesis nor produced pollen. Intriguingly, we observed that flowers that opened 8, 9, and 10 DARW were completely male sterile as pollen was not shed from these anthers and therefore recorded zero percent pollen viability. The flowers that opened on 11 DARW had a mixture of both perfect sterile and fertile anthers and recorded 48.3% pollen viability, which was significantly lower than in the WW plants. From 12 to 19 DARW, pollen shedding and viability (98.1%) were restored which was not significantly different from that in WW plants (Figure 2B,C). Interestingly, the completely sterile anthers obtained from the flowers that opened from 8–11 DARW, had their origin from meiotic stage drought-stressed floral buds whereas the anthers with male fertility fully restored 11 DARW and above, had their origin from pollen mother cell stage drought-stressed floral buds. This indicates that although meiotic (MEI) and pollen mother cell (PMC) floral buds tolerated the drought stress, PMC floral buds appeared to be more drought tolerant. Our results suggested that floral buds spanning the period from the meiotic stage (−11 DAA) TPM1 (−5 DAA), the stage between VUM and BIN (Appendix A) when the nucleus in the developing pollen starts its first mitotic division, exhibited severe abnormalities in anther/pollen development under drought stress and might have mainly contributed to the drastic reduction in fruit yield of the DS tomato.

To confirm our speculation that abnormalities in the male organ development due to DS might have crucial roles in the poor fruit set/yield of the DS plants, pollination experiments were performed to evaluate fruit set and flower abortion among the following groups of plants: (1) self-pollinated well-watered (WW) plants (WW **⊗**); (2) self-pollinated drought-stressed (DS) plants (DS **⊗**); (3) stigmas of flowers on WW plants cross-pollinated with pollens from DS plants (WW 
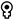
 × DS 
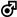
); and (4) stigmas of flowers on DS plants cross-pollinated with pollen grains from WW plants (DS 
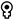
 × WW **
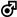
)**. When the stigmas of flowers on WW plants were pollinated with pollen of flowers from DS plants (WW 
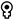
 × DS 
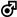
), fruit set was 55.5% lower than in WW plants, which was consistent with a higher flower abortion rate (Figure 2D), indicating that DS severely affected the male fertility. In contrast, when the stigmas of flowers on DS plants were pollinated with sound pollen of flowers from WW plants (DS 
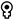
 × WW 
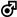
), fruit set and flower abortion were 85.7% and 14.3% which were moderately higher and lower respectively than those in WW plants (Figure 2D,E), indicating that the female fertility was not significantly affected. These results indicated that drought stress-induced irregularities in the male organ is the main cause of floral bud and open flower abortions and the ensuing low fruit set/yield of drought-stressed tomato plants.

### 3.3. Histo-Cytological Analysis of Anther Development under Drought Stress

To understand details of the effects of drought stress on the male development, cross-sections of anthers at seven developmental stages including pollen mother cell (PMC), Meiotic (MEI), tetrad (TED), early uninucleate microspore (EUM) vacuolated uninucleate microspore (VUM), binucleate (BIN) and mature pollen (MP) adapted from [27,38] were examined under a light microscope (LM) and transmission electron microscope (TEM). 

Histologically, pre-meiotic pollen mother cell (PMC) anthers from DS plants exhibited no distinct alterations from WW anthers (Appendix A). However, defects were obvious at all of the later stages examined after 4 days of DS treatment (Figure 3A). From MEI to EUM stage, the tapetum layer was dwindled and completely degenerated at the VUM stage. At the MEI stage, the sporogenous tissue was constricted and disengaged from the anther wall especially in the outer tapetum region (arrow 1) so that the sporogenous tissue no longer filled the locular space. At the TED stage, the callose wall had ectopically dissolved and released the microspores precociously. At the EUM stage, the microspores were compressed and compactly packed. At the VUM abnormalities included degenerated pollen (arrows 2) and premature induction of anther dehiscence (arrow 3). At the BIN stage, pollen grains were degenerated (arrow 2) and enlarged (arrow 4). At the MP stage, the stomia cells partially degenerated (arrow 5) and a large number of pollen grains were retained in the anther locules probably due to inefficient dehiscence.

The effects of DS on pollen development were further revealed by TEM observations (Figure 3B and Appendix A). Developing pollen cells at different stages of development were plasmolyzed with constricted protoplasm separated from the pollen cell wall (arrow 6) and had abnormally numerous small or large vacuoles (Figure 3B). Other abnormities observed in the developing pollen included expansion of the middle lamella at the MEI stage and premature microspore wall formation at the TED stage (Appendix A).

In addition, TEM observations revealed disturbances in tapetum development caused by drought stress (Figure 3C). Under well-watered condition, the tapetum degeneration occurred at VUM stage. However, in DS anthers from MEI to EUM stage, the tapetum underwent ectopic degeneration with constricted protoplasm at MEI stage (arrow 8) and completely devoid of protoplasmic contents and cellular structure at TED and EUM stages respectively. In short, drought stress spanning the period from meiotic to the binucleate stage can adversely affect anther and pollen development through induction of tapetum/pollen degeneration and dissolution, and stomia persistence/anther indehiscence while the pollen wall development is less affected by drought stress.

### 3.4. Transcriptome Changes in Tomato Anthers under Drought Stress

To study possible molecular modifications in tomato anthers under drought, Illumina RNA Sequencing (RNA-Seq) was used to investigate the transcriptome of three groups of anthers: PMC-MEI, TED-VUM and BIN-MP from WW- and DS-plants (Figure 1C). After filtering, a total of 924,197,894 clean reads were generated from 18 libraries with high consistency between replicates (Pearson’s *r* = 0.86–0.99, Appendix A). Out of that, 862,078,735 reads, uniquely aligned to specific genomic regions of *S. lycopersicum* reference genome with an average mapping rate of 98.2% (Appendix A). In total, 15,431 DEGs were identified (Appendix A). Among them, 7022 were up-regulated: 3427, 2780, and 815 genes in the PMC-MEI, TED-VUM, and BIN-MP anthers, respectively (Figure 4A and Appendix A), whilst 8409 were down-regulated: 3450, 3560 and 1399 genes in the PMC-MEI, TED-VUM and BIN-MP anthers, respectively (Figure 4B and Appendix A). This indicates that the number of DEGs decreased with the age of anthers, more genes were down-regulated than up-regulated, PMC-MEI anthers had the highest number of DEGs and a greater number of genes was down-regulated in the TED-VUM anthers indicative of their high level of vulnerability to drought stress. It is worth pointing out that although there were many DEGs at different stages, only 412 were constitutively up-regulated (128) and down-regulated (284) at different developmental stages indicating that at different stages of flower development, different genes may respond to drought stress. Confirmation of RNA-Seq results with RT–qPCR analysis using 18 selected drought-responsive genes revealed that the expression patterns from RT–qPCR analysis were consistent with those from RNA-Seq analysis (Appendix A).

To have an idea of the possible roles of the DEGs, enrichment analyses of GO terms and KEGG pathways were performed. A total of 51 functional groups were significantly enriched in up-regulated DEGs (Appendix A) whereas 174 functional groups were significantly enriched in down-regulated DEGs (Appendix A). For biological processes, the most enriched GO terms were significantly enriched in down-regulated genes in more than one sample such as translation (GO: 0006412), organonitrogen compound metabolic process (GO: 1901564) and amide biosynthetic process (GO: 0043604) in PMC-MEI and BIN-MP anthers; metabolic process (GO: 0008152) in PMC-MEI and TED-VUM anthers. Other functional groups were only enriched in one specific sample, for example, carbohydrate metabolic process (GO: 0005975) and lipid metabolic process (GO: 0006629) in TED-VUM anthers. This suggested that genes associated with carbohydrate and lipid-derived hormones such as jasmonic acid metabolisms were repressed by drought stress in TED-VUM anthers. Similarly, KEGG analysis showed only one pathway ‘Ribosome’ highly significantly enriched in down-regulated DEGs in PMC-MEI and BIN-MP anthers whereas others were only enriched in TED-VUM anthers, such as “photosynthesis”, “biosynthesis of secondary metabolites” and “metabolic pathways” (Figure 4B and Appendix A). Strikingly, only one pathway, “plant hormone signal transduction” was highly enriched in up-regulated DEGs in PMC-MEI anthers (Figure 4B and Appendix A). Together, our GO and KEGG data suggest that drought stress may primarily affect carbohydrate and secondary metabolic processes including hormone pathways in tomato anthers.

### 3.5. Impact of Drought Stress on Tapetum-Specific Expressed Genes

To determine whether drought stress affected tapetum and microspore development at the transcription level, we compiled a list of 52 tapetum- and pollen-related genes [39,40,41,42,43,44,45] (Appendix A). RNA-Seq data showed that *S. lycopersicum Excess Microsporocytes1* (*SlEMS1*), *Aborted Microspores* (*SlAMS*), *Defective in Tapetum Development and Function1* (*SlTDF1), Dysfunctional Tapetum1 (SlDYT1)* and *Callose Defective Microspore1* (*SlCDM1*) were markedly down-regulated whereas *SlMYB80* was up-regulated under DS condition in PMC-MEI anthers. In the TED-VUM anthers, *SlAMS*, *Male Sterility1* (*SlMS1*) and *Anther7* (*SlATA7)* were significantly drought-repressed (Figure 5). These genes were shown to play important roles in tapetum/pollen development [46,47,48,49]. The up-regulation of *SlMYB80* in PMC-MEI anthers is consistent with the normal development of the PMC anthers that survived the DS, developed to anthesis and produced viable pollen indicating normal pollen and tapetum development after rewatering. On the other hand, the down-regulation of all examined tapetum-related genes including *SlMYB80* in TED-VUM anthers is in line with the considerable amount of damage observed in the tapetum and/or microspores in TED-VUM anthers as revealed by histo-cytological analysis. These results suggest a role for SlMYB80 in drought resistance mechanisms during the early stages of anther development under drought stress.

### 3.6. Drought Stress Disturbs Sucrose and Starch Metabolism and Soluble Sugar Transport

Consistent with “carbohydrate metabolic process” and “photosynthesis” pathways significantly enriched in drought-regulated genes, 16 DEGs involved in sucrose cleavage, sugar phosphorylation and starch synthesis, and 33 DEGs associated with sugar transport were identified (Appendix A). Gene expression analysis showed that the sucrose cleavage genes including *cell wall invertase 3-like (SlCWINV3-like)* and *fruit sucrose synthase* (*TOMSSF*) were significantly repressed in PMC-MEI anthers. Whereas the genes encoding *β-fructofuranosidase* and *SlSUS6* were significantly up-regulated in TED-VUM anthers, *SlCWINV3-like* and *SlSUS6* were significantly repressed in BIN-MP anthers (Figure 6A), suggesting that sucrose hydrolysis is impaired in both PMC-MEI and BIN-MP anthers but not in TED-VUM anthers under drought stress. The sugar transporters, *SlSWEET16* and *SlSTP8* were notably up-regulated in PMC-MEI anthers. *SlSWEET16* was markedly induced at both TED-VUM and BIN-MP anthers (Figure 6A), suggesting that sugar transport is generally not inhibited in tomato anthers under drought stress. Further utilization of hexose sugars in metabolic processes requires their phosphorylation, catalyzed by hexokinases that phosphorylate both glucose and fructose, and fructokinases that specifically phosphorylate fructose [50]. While *Slhxk1* and *Slhxk2* were both significantly repressed in MC-MEI anthers, only *Slhxk1* was significantly down-regulated in TED-VUM anthers. In addition, both the large and small subunits of the rate-limiting enzyme in starch biosynthesis, *Slagpl3* and *TOMADPGPPS,* respectively, were significantly down-regulated in PMC-MEI and TED-VUM anthers. However, the gene encoding the starch hydrolyzing enzyme, β-amylase 8, was significantly up-regulated in BIN-MP anthers (Figure 6A). Our data suggest that while drought might hinder anther sink strength and starch biosynthesis, it might enhance starch degradation during pollen development.

Under DS conditions, sucrose, fructose and glucose contents were strikingly higher in PMC-MEI and BIN-MP anthers, although glucose was significantly lower in PMC-MEI anthers whereas their contents remained similar to those in WW anthers in TED-VUM anthers (Figure 6B). In drought-stressed anthers, the pattern of starch accumulation in developing pollen was comparable to that in WW anthers from MEI to VUM stage. However, in BIN-MP anthers, starch accumulation in the pollen grains of DS anthers was attenuated, whereas more starch accumulated in the pollen grains under WW conditions (Figure 6C).

### 3.7. Effect of Drought on Phytohormones Metabolism and Signaling

The substantial number of DEGs involved in diverse plant hormones metabolism and signaling found in the transcriptome, fascinated us to explore the contribution of IAA, ABA and JA in drought responses of tomato anthers (Figure 7A). Furthermore, the expression profiles of genes involved in IAA, ABA and JA metabolic and signaling pathways based on RNA-Seq data and the corresponding endogenous hormone contents were analyzed. A total of 38 DEGs encoding enzymes involved in auxin metabolism (17) and signaling (21) were at least regulated in one group of anthers (Appendix A). The IAA biosynthesis gene, *S. lycopersicum tryptophan*
*aminotransferase of Arabidopsis*-*related* 2 (*SlTAR2),* was significantly down-regulated in PMC-MEI anthers, whereas *SlTAR1-like*, *SlTAR*2, *FLOOZY1* (*ToFZY1*) and *ToFZY2* were significantly up-regulated in TED-VUM anthers (Figure 7B). Consistently, the content of endogenous IAA was significantly reduced in PMC-MEI anthers whereas it was moderately increased in TED-VUM anthers under DS condition (Figure 7C).

A total of 19 DEGs associated with ABA metabolism and signaling were identified (Appendix A). In the PMC-MEI anthers, *Slcyp707a1,* encoding ABA 8′-hydroxylase involved in ABA catabolism, was markedly up-regulated. The core components of ABA signaling genes including one ABA receptor *SlPYL9*, two protein phosphatase 2C (PP2C) genes, *ABA insensitive 2* (*SlABI2*) and *ABA-hypersensitive germination 3* (*SlAHG3*) [51,52], and two *SNF1-Related Protein Kinase2* (*SnRK2*) genes (*SlSRK2C* and *AY222455*) were likewise significantly up-regulated (Figure 7B). Additionally, two known direct downstream targets of *SnRK2, SlAREB2* (homologue of *AtAREB2*) and *SlABF3* (homologue of *AtABF3*) (Appendix A), were exclusively significantly up-regulated in PMC-MEI anthers, suggesting that ABA signaling is induced early during another development under drought stress. In TED-VUM anthers, the expression levels of all examined genes associated with ABA were not significantly different from those in WW anthers suggesting ABA signaling might have been impaired. However, in the BIN-MP anthers, some ABA biosynthesis and catabolic genes, *SlSDR* (homolog of *AtSDR2*) [53] and *Slcyp707a1*, respectively, and some signaling genes including *SlABI2, SlAHG3* and *SlSRK2C* genes were moderately up-regulated (Figure 7B). This suggests that ABA biosynthesis and catabolism are moderately increased but signaling impaired due to lack of inefficient expression of ABA receptors under drought conditions in BIN-MP anthers. Although ABA biosynthesis genes expression was lower, lower and higher ABA increase in PMC-MEI and TED-VUM anthers respectively (Figure 7C) coincided with marked and very weak induction of ABA catabolic gene in PMC-MEI and TED-VUM anthers, respectively (Figure 7B). The TED-VUM anthers with higher ABA increase succumbed to the DS, whereas the PMC-MEI anthers with lower ABA increase tolerated the DS. Together, these results suggest that regulation of endogenous ABA content through its catabolic pathway might have roles in drought-resistance mechanism in the early stages of anther development.

A total of 21 DEGs associated with JA metabolism and signaling were recognized in at least one group of anthers (Appendix A). RNA-Seq analysis of four genes involved in JA biosynthesis revealed that in PMC-MEI anthers, *SlLOX5* exhibited significantly increased expression. In TED-VUM anthers, all four genes: *SlLOX5*, *SlOPR3*, *Phospholipase A2* and *SlAOC* were significantly down-regulated under drought stress and likewise in BIN-MP anthers although none was significant (Figure 7B). Invariably, under DS, endogenous JA content was significantly decreased in TED-VUM and BIN-MP anthers while it was increased although not significantly in PMC-MEI anthers (Figure 7C). These results suggest a probable role of JA in the very early (PMC and microsporocyte) stages of anther development under drought stress.

## 4. Discussion

Successful anther development in flowering plants is very crucial to ensuring plant fertility and productivity because it produces and delivers the male gamete to the female gametophyte for efficient fertilization [54]. However, anther development is often perturbed by abiotic stresses including drought resulting in male sterility and yield reduction [17,55]. Nevertheless, the developmental flaws and the underlying physiological and molecular mechanisms remain unclear in tomatoes. This study, examined the effect of drought stress on anther development using morpho-physiological and molecular analyses in tomatoes. Drought stress affected anther/pollen development in an age-dependent manner. The drought induced arrest of anther/pollen development and caused irreversible growth inhibition (abortion) of anthers at tetrad and uninucleate microspore stages (Appendix A), which reduced the number of flowers reaching maturity (Figure 1B) consistent with [56]. Previous studies attributed yield reduction to decreased pollen viability and increased opened flower abortion emanating from reduced and/or absence of pollen starch accumulation [55,57]. In line with these observations, drought stress (DS) in our study resulted in reduced/and or absence of pollen starch accumulation and male sterility in mature pollen (BIN-MP) (Figure 6C). Although the floral buds in the PMC-MEI group endured the drought stress to a certain level (Appendix A), the anthers in the MEI floral buds were susceptible to the drought stress which exhibited abnormal development of the anther structure, pollen and tapetum cells and pollen sterility (Figure 3A–C), resulting in total lack of pollen shedding at maturity 8–10 days after rewatering (Figure 2C). A similar observation was made in the meiotic stage drought-stressed wheat anthers [58]. We further demonstrated that pollen sterility is a major but not the only yield reduction factor under DS. Altered anther dehiscence in MP anther (Figure 3A) and stigma exertion in flowers derived from meiotic stage drought-survived anthers (Figure 1D and Appendix A), reported in chickpea and tomato under low and high-temperature stresses, respectively [25,59], contributed greatly to opened flower abortion and subsequently yield reduction in our study. The inherent ability of young PMC-MEI anther to resist drought and resumed growth after rewatering led to a delay in flowering for about 3 days and bimodal flower production in DS tomato plants (Figure 1B and Appendix A). Our results suggest that drought reduction in fruit yield is due mainly to pollen infertility but secondary factors including anther indehiscence and stigma exertion also play roles. Importantly, the identification of diverse sensitivities to drought stress, resistant (PMC) and sensitive (TED-VUM) stages in our study is a significant breakthrough to identifying major players involved in drought stress tolerance that can facilitate future drought tolerance breeding in tomatoes.

Normal tapetum function, regulated by coordinated expression of tapetum expressed genes, is a prerequisite for the production of viable pollen [60]. Alteration in tapetum development leads to male sterility. This is supported by the observation that DS anthers from meiotic to early uninucleate microspore stage exhibited abnormal tapetum development (Figure 3A,C). This occurred concurrently with altered expression of tapetum-expressed genes (Figure 5) in agreement with previous reports [17,61]. Although DS interference with tapetum development generally results in pollen sterility, we showed that drought-induced alterations in tapetum development varied with the developmental stage. Drought ectopically degenerated the tapetum in tetrad and early uninucleate microspore anthers and resulted in floral bud abortion (Figure 1D,E and Appendix A) with conspicuous down-regulation of all examined tapetum expressed genes (Figure 5). In drought-survived PMC-MEI anthers, SlMYB80 was markedly induced and a set of anthers within the PMC-MEI group, PMC anthers that reached anthesis 12 days after rewatering (DARW) and afterward, exhibited normal anther/pollen development, demonstrated by production of viable pollen after rewatering (Figure 2B,C). This suggests a role for *SlMYB80* in the early stages of anther development under drought stress and a possible target for engineering drought resistance in tomatoes.

Starch forms the principal storage food reserve in mature pollen [17]. Arriving sucrose in the anther is hydrolyzed by invertases into monosaccharides, transported into cells through monosaccharide transporters and used in starch biosynthesis [62]. Studies have reported that drought stress at reproductive stage alter sucrose and starch metabolisms and cause male sterility through a reduction in pollen fertility [63]. Our study showed that sucrose, glucose, fructose, and starch accumulated abnormally in developing drought-stressed anthers (Figure 6), suggesting that drought affects carbon allocation and processes implicated in starch biosynthesis. In the drought-resistant PMC-MEI and BIN-MP anthers, *SlCWINV3-like* and *TOMSSF*, and *SlSUS6* were down-regulated respectively, which might have caused higher sucrose levels in agreement with previous reports [64]. This eliminates deficit carbohydrate supply as a possible cause of reproductive failure in these anthers. In contrast, the induction of *β-fructofuranosidase* and *SlSUS6* in the drought-sensitive TED-UM anthers, correlated with lower sucrose levels which might be associated with deficit carbohydrate supply as a probable cause of reproductive failure in the TED-VUM anthers. A similar observation described previously [14], demonstrated that soluble sugars accumulated more in anthers of drought-tolerant than in drought-sensitive genotype.

Committing hexose sugars from sucrose cleavage to participate in starch biosynthesis and other metabolic processes requires their phosphorylation by hexokinase, demonstrated to be specifically expressed in the tapetum and pollen of developing anther [50,65]. In this study, we observed two hexokinase genes, *Slhxk1* and *Slhxk2* significantly repressed in PMC-MEI anthers (Figure 6A) resulting in an increased level of the monosaccharide sugar, fructose in agreement with [64]. However, glucose was decreased in the PMC-MEI anthers suggesting its continued basal level utilization as an energy source to fuel life-supporting biological processes during the DS, demonstrated by the resistance to drought stress and normal development of the PMC anthers after rewatering (Appendix A). In addition, the expression of the *ADGPase* gene was repressed by drought stress consistent with [18] while *β-amylase8*, was induced at the BIN-MP stage. These results suggest that the poor reproductive performance of tomatoes under drought stress can be ascribed to deficit carbohydrate supply in the sensitive anthers, and diminished sugar utilization and hydrolysis of existing starch in maturing and drought-resistant anthers.

Previous studies provided evidence of IAA involvement in abiotic stress responses in vegetative organs [66,67]. We demonstrated that drought-repression of *SlTAR2* in PMC-MEI anthers and induction of *SlTAR2* and *SlTAR1-like*, *ToFZY1* and *ToFZY2* in TED-VUM anthers (Figure 7B), were in agreement with reduced and increased IAA levels in these anthers, respectively (Figure 7C). Intriguingly, drought-stressed TED-VUM anthers exhibited severe pollen abnormalities (Figure 3) and aborted (Figure 1D and Appendix A) while PMC-MEI anthers survived and a subset of flowers obtained from them, produced viable pollen (Figure 2B,C) inconsistent with [22], who reported that decreased IAA biosynthesis genes expression and contents cause decreased pollen and spikelet fertility in rice. Our results suggest a negative interaction between IAA with sugar signaling during anther development under drought stress in tomatoes.

Drought stress triggers ABA biosynthesis and increases its content in reproductive organs with a negative correlation shown to exist between ABA contents and abiotic stress tolerance [19]. Expression analysis revealed that *SlSDR*, a homolog of *AtSDR2*, was significantly induced in BIN-MP anthers, *SlNCED1* the key ABA biosynthesis gene [68] and *SlSDR*, were induced in PMC-MEI and TED-VUM anthers though not significantly while the ABA catabolic gene, *Slcyp707a,* was highly and moderately induced in PMC-MEI and BIN-MP anthers, respectively (Figure 7B). Although ABA biosynthesis genes expression in the PMC-MEI (drought resistant) and TED–VUM (drought-sensitive) anthers was insignificant, ABA level was increased by 205% and 395% respectively. This indicates that the marked and insufficient induction of Slcyp707a is the cause of low and high ABA levels in the PMC-MEI and TED-VUM anthers, respectively, in line with [20,69,70] suggesting that Slcyp707a plays important roles in the drought resistance mechanism in the early stages of anther development. Put together, under drought stress, our results suggest the ABA catabolic pathway as a decisive player in regulating ABA homeostasis and drought resistance in tomatoes and a negative interaction between IAA and ABA in the early stages of anther development.

To sum up, the increased anther and opened flower abortions and subsequent reduction in fruit yield of DS plants can be attributed to drought induction of male sterility caused by abnormalities in anther and pollen development. Under drought stress, anthers exhibited differential sensitivities to DS. Sensitivity of anthers to drought stress spanned the period from meiotic mother cell stage to binucleate stage with TED-VUM anthers, the most sensitive to drought and PMC anthers, most insensitive to drought. The drought resistance exhibited by the PMC-MEI anthers can be associated with a moderate increase in ABA level due to the high-level expression of its catabolic gene that maintains the level of ABA optimum to trigger signaling and activation of ABA-dependent drought adaptive gene expression and repression of IAA signaling (Figure 8). Our findings provide insight into the behavioral patterns and defects in the anthers and pollen of different stages of development and the associated physiological and molecular mechanisms under DS and provide a novel insight into potential drought tolerance mechanisms which can be engineered for improvement of drought tolerance in tomatoes.

## Figures and Tables

**Figure 1 cells-10-01809-f001:**
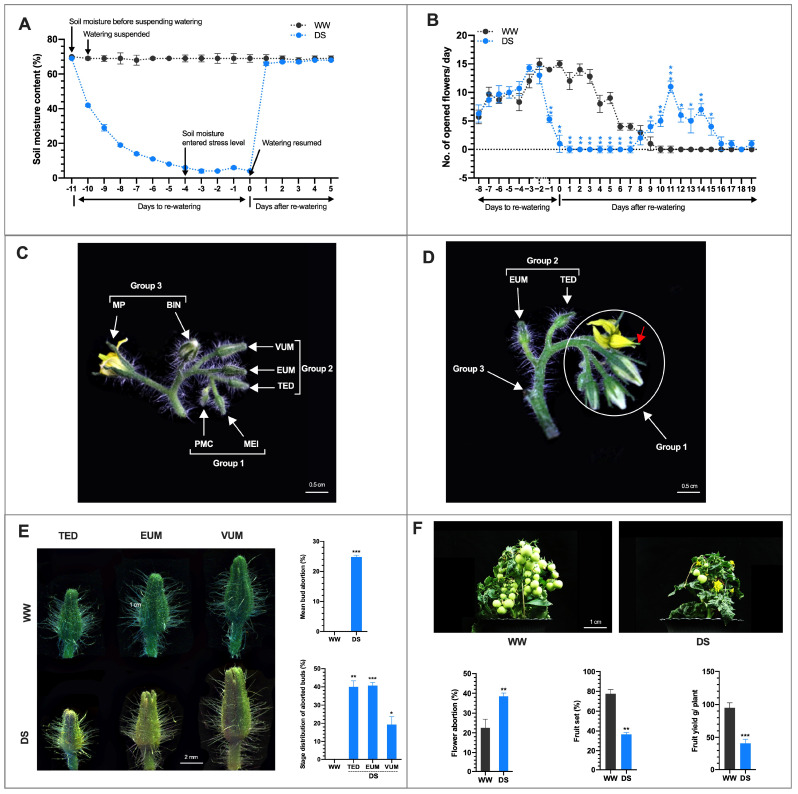
Tomato anther development and fruit yield under drought stress. (**A**) Soil moisture content (%) during the period watering was suspended to 5 days after re-watering (DARW). Data points and error bars represent means (*n* = 16). (**B**) Number of opened flowers/day before and after rewatering. Data points represent mean ±SD (*n* > 10). (**C**) A truss from DS plant at 0 DTRW with one newly opened flower and an array of unopened floral buds at different stages of development. The BIN floral bud was −1 DAA. (**D**) A DS truss similar to C above, 9 DARW. The BIN (−1 DAA) floral bud that opened I DARW had its opened flower aborted and dropped with only a remnant of the flower stalk left. The group 2 floral buds did not recover from the DS and had aborted. (**E**) Drought stress-induced floral bud abortion at three specific developmental stages namely tetrad (TED), early uninucleate microspore (EUM) and vacuolated uninucleate microspore (VUM). For details see Appendix A. The DS floral buds were declared as ‘aborted’ and sampled on 9 DARW when younger floral buds in group 1 began to flower (Figure 1D). Values are means ±SD (*n* = three biological replications each with aborted floral buds >39). (**F**) Drought caused open flower abortion, reduction in fruit set and fruit yield in tomatoes. Values are means ±SD (*n* = three biological replications, each with >140 opened flowers assessed. * *p* < 0.05; ** *p* < 0.01; *** *p* < 0.001 (*t*-test).

**Figure 2 cells-10-01809-f002:**
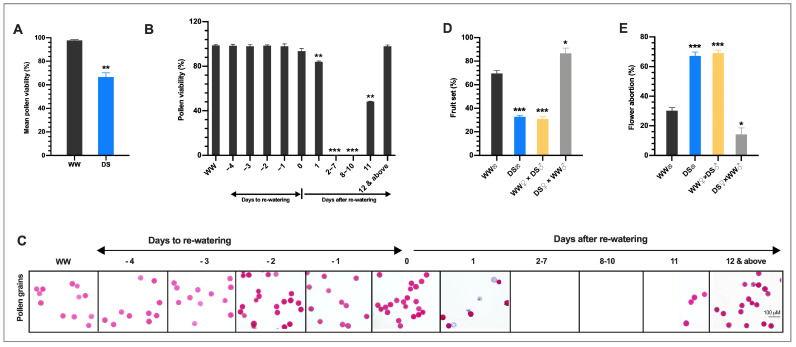
Drought stress reduces the fertility of the male organ. (**A**) Mean pollen viability of all flowers that opened and failed to open on tagged trusses in the DS plants from start to end of flowering. Values presented are means ±SD (*n* = 150) (**B**,**C**). Daily mean pollen viability (**B**) and pollen (**C**) from −4 DTRW to 19 DARW. Data points represent means ±SD (*n* > 10). (**D**) Fruit set (**E**) flower abortion of self-pollinated WW and DS plants, stigma of flowers on WW plants pollinated with pollen grains from DS plants (WW 
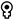
**×** DS 
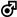
) and stigma of flowers on DS plants pollinated with pollen grains from WW plants (DS 
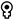

**×**WW 
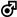
). Values presented are means ±SD (*n* = 150 for WW **⊗** and DS **⊗** experiments; *n* = 125 for WW 
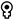
 × DS 
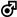
 and DS 
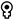

**×** WW 
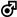
 experiments. * *p* < 0.05; ** *p* < 0.01; *** *p* < 0.001 (*t*-test).

**Figure 3 cells-10-01809-f003:**
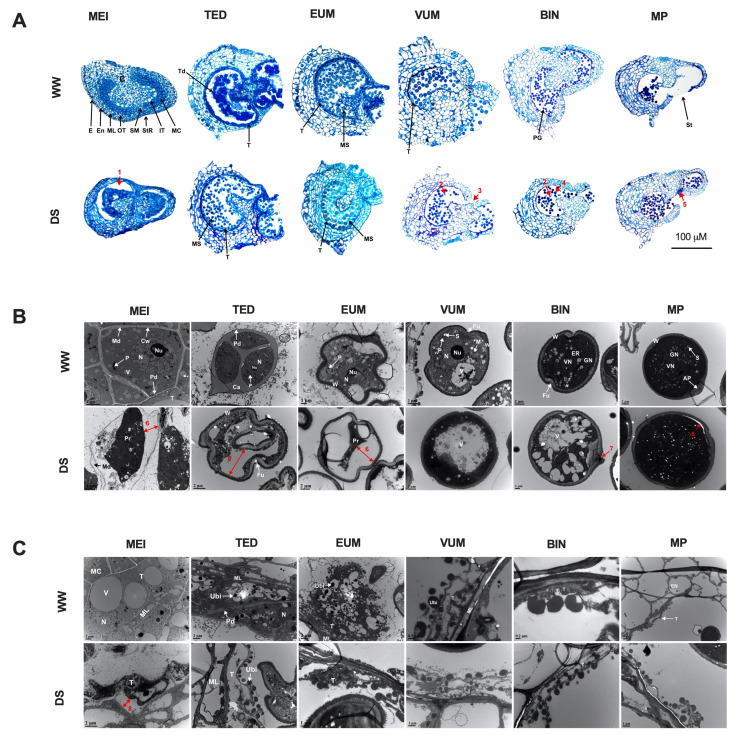
Drought stress affects anther, tapetum and pollen development. (**A**) Cross-sections of anthers at six different stages of development after 4 days under DS. WW, anthers from well-watered plants; DS, anthers from drought-stressed plants MEI, meiotic; TED, tetrad; EUM, early uninucleate microspore; VUM, vacuolated uninucleate microspore; BIN, binucleate; MP, mature pollen are different stages of pollen development; E, epidermis; En, endothecium; ML, middle layer; OT, outer tapetum; SM, septum; StR, stomium region; IT, inner tapetum; MC, meiotic cell; C, connective tissue; Td, tetrad; T, tapetum; MS, microspore; PG, pollen grain; St, stomium. (**B**) Transmission electron micrographs (TEMs) of developing pollen. Cw, cell wall; Nu, nucleolus; N, nucleus; Md, middle lamella; Ca, callose; W, pollen wall; V, vacuole: Fu, furrow; VN, vegetative nucleus; GN, generative nucleus; AP, aperture; S, starch granule. (**C**) TEMs of developing tapetum. ML, middle layer; Ubi, ubisch body; Pd, plasmodesma. Arrows: 1, gap between disengaged sporogenous and tapetum tissues; 2, aborted pollen; 3, precocious anther dehiscence; 4, abnormally enlarged pollen; 5, persistent stomium cells adhered to connective tissue; 6, gap between pollen wall and pollen protoplasm, 7, breakage in pollen wall and 8, gap between tapetum wall and shrank tapetum cell protoplasm due to plasmolysis.

**Figure 4 cells-10-01809-f004:**
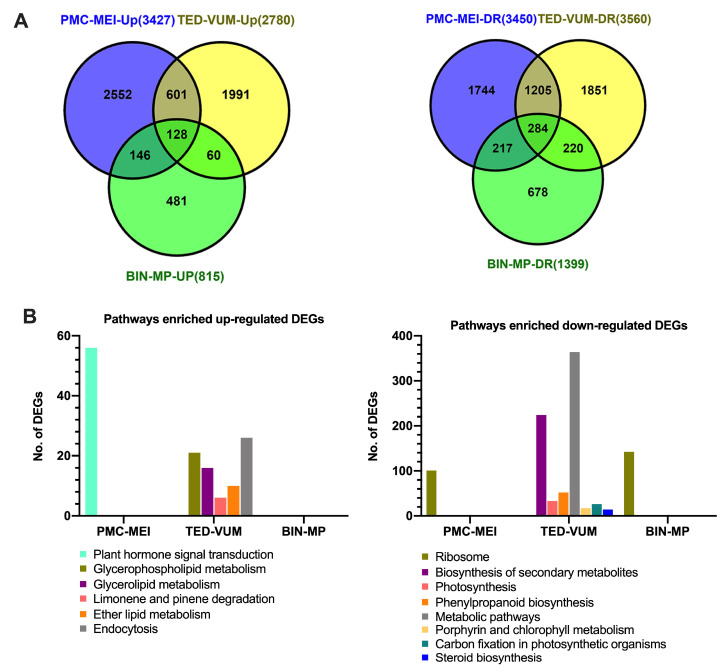
Drought-responsive genes in three groups of anthers of different developmental stages in tomato. (**A**) Venn diagram analyses of DEGs Up-regulated (**left**) and down-regulated (**right**) by drought stress in each group. (**B**) Significantly enriched KEGG pathways in DEGs Up- and down-regulated by drought stress in each group.

**Figure 5 cells-10-01809-f005:**
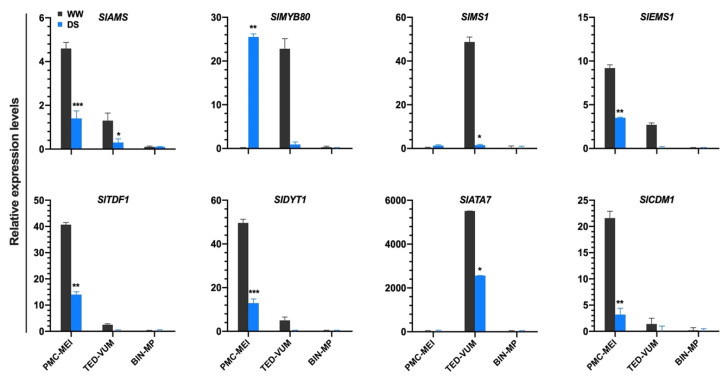
Expression level changes of tapetum-expressed genes in three groups of tomato anthers of different developmental stages under drought stress. Gene expression levels were based on the number of fragments per kilobase of transcript sequence per millions (FPKM) base pairs sequenced from RNA-Seq data and presented as means ±SD (*n* = three replications, 24 plants each). * *p* < 0.05; ** *p* < 0.01; *** *p* < 0.001 (*t*-test).

**Figure 6 cells-10-01809-f006:**
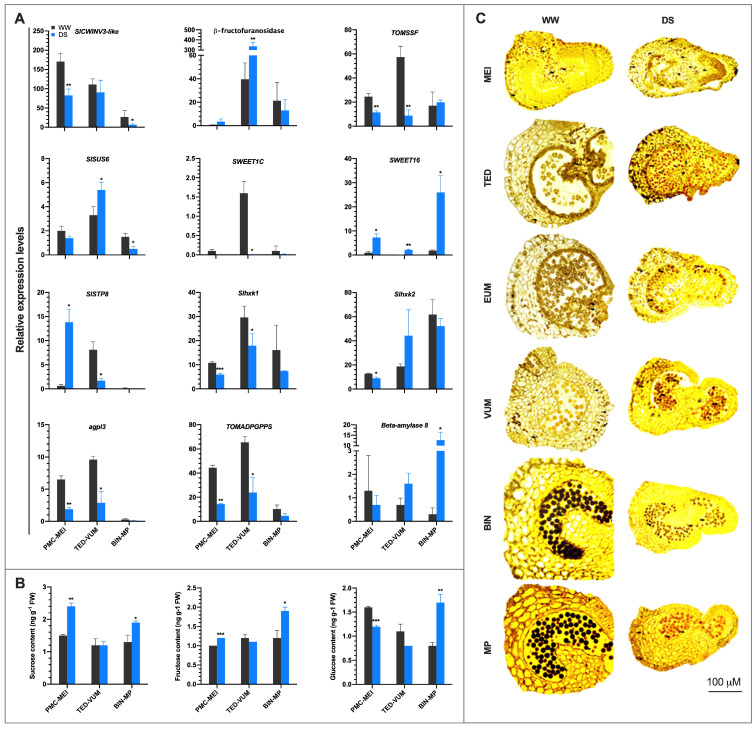
Comparison of sucrose and starch metabolism in three groups of anthers of different developmental stages exposed to drought stress. (**A**) Expression analysis of genes involved in sucrose and starch metabolism, and sugar transport. Gene expression levels were based on the number of fragments per kilobase of transcript sequence per millions (FPKM) base pairs sequenced from RNA-Seq data. (**B**) Sucrose, glucose, and fructose contents in anthers of DS plants. Anthers of equivalent developmental stages were sampled from WW and DS plants 4 days after drought stress. Values are means ±*SD* (*n* = three replications, 24 plants each). * *p* < 0.05; ** *p* < 0.01; *** *p* < 0.001 (*t*-test). (**C**) Starch accumulation in tomato anthers determined by staining with Iodine Potassium Iodide (IKI) solution. WW, well-watered anthers; DS, drought-stressed anthers; MEI, anthers at the meiotic stage; TED, anthers at tetrad stage; EUM, anthers at early uninucleate microspore stage; VUM, anthers at vacuolated uninucleate microspore stage; BIN, anthers binucleate stage; MP, anthers at mature pollen stage.

**Figure 7 cells-10-01809-f007:**
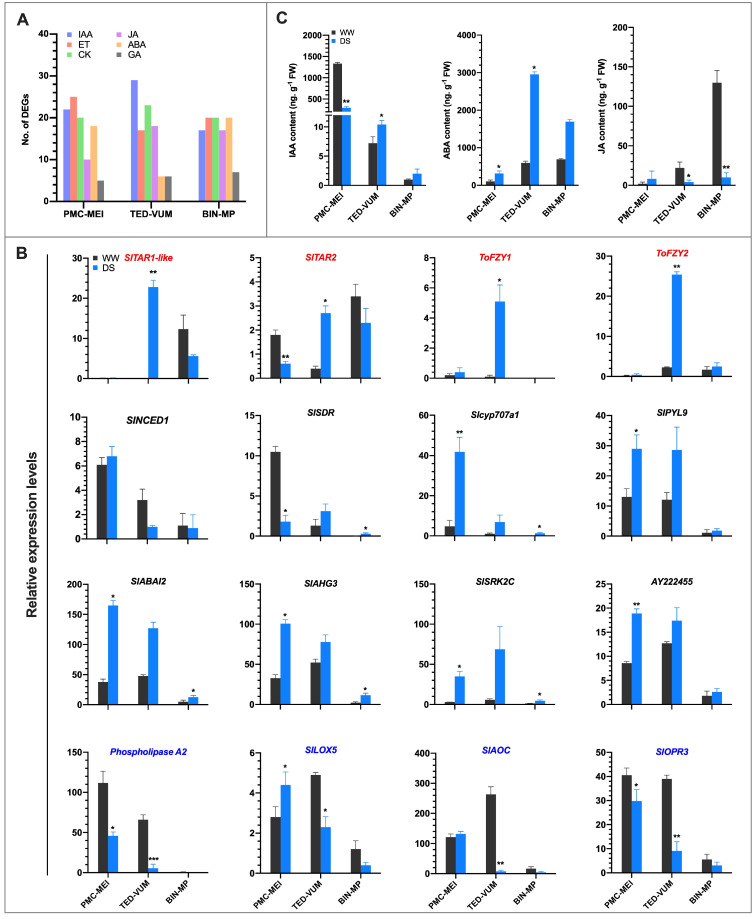
IAA, ABA and JA metabolisms and signaling in three groups of anthers of different developmental stages under drought stress in tomato. (**A**) Distribution (%) of different phytohormones metabolic and signaling DEGs in tomato anthers. (**B**) Expression analysis of genes involved in IAA (red titles), ABA (black titles) and JA (blue titles) metabolisms and/or signaling in DS anther. Gene expression levels were based on the number of fragments per kilobase of transcript sequence per millions (FPKM) base pairs sequenced from RNA-Seq data. (**C**) Contents of IAA, ABA and JA in anthers 4 days after DS. Values are means ±SD (*n* = three replications, 24 plants each). * *p* < 0.05; ** *p* < 0.01; *** *p* < 0.001 (*t*-test).

**Figure 8 cells-10-01809-f008:**
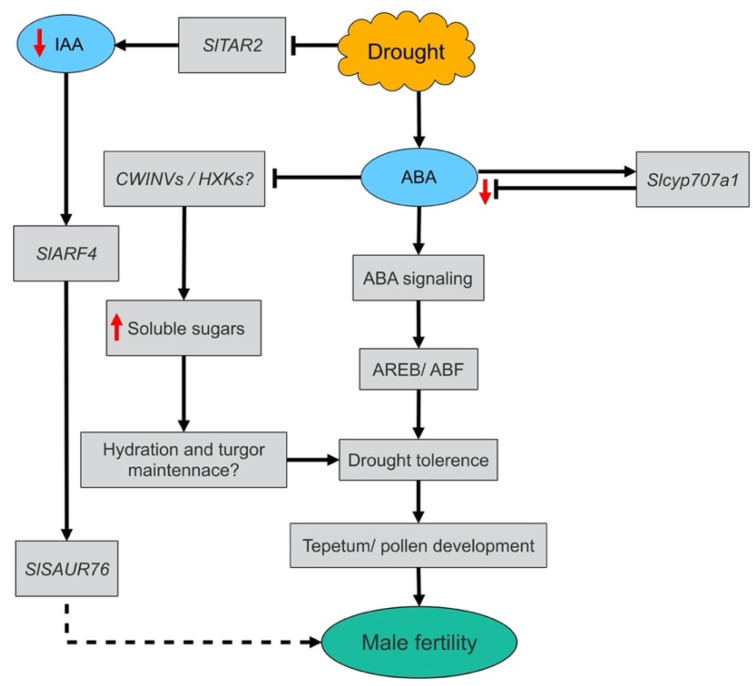
Proposed model of drought tolerance mechanisms in tomato anthers during early stages of development. ABA: abscisic acid; IAA: indole-3-acetic acid; CWINVs: cell wall invertases; HXKs: hexokinases; AREB: abscisic acid-responsive element-binding protein; ABF: abscisic acid-responsive element binding factor; Up-ward pointing red arrow: significant increase; Down-ward pointing red arrows: significant decrease; Dash line: suggested to play a role, early in anther/pollen development since SlAUR76 (*ID: Solyco8g077020.1*) was significantly repressed in our transcriptome but has not been functionally proven.

## Data Availability

All datasets generated for this study are included in the article/Appendix A.

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
