# Peer review of "Morpho-Physiological and Transcriptome Changes in Tomato Anthers of Different Developmental Stages under Drought Stress"

_cells, 2021, doi:10.3390/cells10071809_

Round 1
Reviewer 1 Report
The authors characterized tomato buds/anthers/pollen development under drought stress from the aspects of morphology, physiology, and transcriptome. Male development in plants including tomato is generally quite sensitive to climate conditions, which resulted in poor yield performance, thus the mechanisms of the effect of and tolerance against stresses have received high attention. The drought responses in anther/pollen development have been characterized in some species, e.g. rice, while those in tomato have not been enough documented. The authors discussed the obtained results with comparative view with those in other species and presented both consistent and unique responses in tomato. This study in my view presented important aspects associated with morphology, carbohydrate metabolism, and hormone metabolism, potentially leading to further understanding and improvement of drought resistance in tomato, although the manuscript needs minor improvement to reach a standard for publication.
Major remarks
1. The authors showed that anthers and buds of PMC were resistant against DS, while anthers, not buds from the view of flower opening, of MEI were susceptible resulting in abnormal development of the anther structure, pollen cells, and tapetum cells, and pollen sterility. While the authors might carefully treated this in the discussion of the latter part, this was little complicated and should be properly clarified again in discussion, maybe in the first paragraph, to avoid confusing readers.
2 - Page 4 line 183.
The definition for the three groups repeatedly described in the results but not only for RNA-Seq. This definition should be clarified before, e.g. in section 2.4, and may be briefly mentioned in the results section 3.1, not associated with RNA-Seq. Besides, the group 1 anthers in Fig. 1D were obviously out of the definition explained here, so it should be addressed.
3 - Page 19 line 668.
It was not clear whether the authors described the associations of hexokinases with the contents of hexoses in PMC-MEI stage. Hexokinases are generally though to more function for phosphorylation of glucose in tomato fruit. The content of glucose was decreased in the PMC-MEI anther and seems not increased even for the total amount of fructose and glucose.
4 -Figure 2D, E.
These data seemed redundant, since the rate of unset fruit might have the same meaning to the rate of flower abortion. How were these parameters, particularly fruit set, measured?
5 - Page 6 line 290-291.
I could not understand what did the following sentence, particularly the numbers, mean: "4" of them reaching anthesis during the DS period while "1" remained unopened... The authors should rephrase the sentence.
6 - Figure 5, 6, and 7.
The authors should clarify the calculation procedures for the "Relative expression levels" in the legends and/or the materials and methods, because it must be based on the RNA-Seq data.
Did they mean FPKM?
7 - Page 13 line 480.
The authors referred here to "lipid-derived hormone metabolisms" repressed by drought, but it was not clear from the descriptions shown. The associated genes with hormones (JA?) in "lipid metabolic process" may support the suggestion.
Minor remarks
- Figure 1A
The label "sampling time" indicated by an arrow is quite confusing, which may indicate sampling for RNA-Seq and not for data of this figure. The label should be removed or more properly explained.
- Figure 1E
There is no scale bar.
- Figure 1E
The authors should describe the timing of plant stage/age in the pictures.
-Figure 2D
The labels for crossing, male x male, need to be corrected.
- Figure 3A
I would suggest to label the tapetum also in the EUM of WW.
- Figure 8
The explanations for red arrows and dashed line are needed.
- Page 3 line 123
This is the first place the abbreviation of drought stress, DS, appeared, thus it should be spelled out here. Please confirm the similar cases through the manuscript. For example, abbreviations of stage names, e.g. PMC, TED etc., and DARW were repeatedly spelled out and it reduced readability.
- Page 4 line 181
"RNA-Sequence" should be "RNA-Sequencing"
- Page 4 line 185
"group I" should be "group 1"
- Page 6 line 254
typo "ddecrease"
- Page 6 line 257
For the preposition of DTRW and DARW, "on" sounds uncomfortable and "at" may be more suitable. Please confirm.
- Page 6 line 266
"1 day after rewatering" should be "One day after rewatering"
- Page 7 line 316
typo "ADRW"
- Page 9 line 349
The abbreviation of DAA should be days after anthesis.
- Page 9 line 370
Brief explanation of pollen mitosis 1 here or in the materials and methods.
- Page 9 line 382-386
The citation positions for Figure 2D and E should be corrected.
Author Response
Reviewer 1
- The authors showed that anthers and buds of PMC were resistant against DS, while anthers, not buds from the view of flower opening, of MEI were susceptible . While the authors might carefully treated this in the discussion of the latter part, this was little complicated and should be properly clarified again in discussion, maybe in the first paragraph, to avoid confusing readers.
Author’s response: Thank you very much for an excellent suggestion. The authors carefully discussed the susceptibility of MEI anthers to drought stress in the discussion section in accordance to your suggestion. See page 18-19 line 637-642.
Below lines are added in revised manuscript:
Although the floral buds in PMC-MEI group endured the drought stress to a certain degree (Figure S2D), the anthers in the MEI floral buds were susceptible to the drought stress which exhibited abnormal development of the anther structure, pollen and tapetum cells and pollen sterility (Figure 3A-C), resulting in total lack of pollen shedding at maturity 8-10 d after rewatering (Figure 2C). A similar observation was made in meiotic stage-drought stressed wheat anthers [58].
2 - Page 4 line 183.
The definition for the three groups repeatedly described in the results but not only for RNA-Seq. This definition should be clarified before, e.g. in section 2.4, and may be briefly mentioned in the results section 3.1, not associated with RNA-Seq. Besides, the group 1 anthers in Fig. 1D were obviously out of the definition explained here, so it should be addressed.
Author’s response: Thank you very much for the salient suggestions. The authors followed your advice and moved the definition for the three groups of anthers from section 2.5 .1 to section 2.4. See page 5 line 173-178; the three groups of anthers briefly mentioned in result section 3.1 and the issue of group 1 anthers in Figure 1D was already addressed by citing Figure 1C in result section 3.1. See page 12 line 457-458.
3 - Page 19 line 668.
It was not clear whether the authors described the associations of hexokinases with the contents of hexoses in PMC-MEI stage. Hexokinases are generally though to more function for phosphorylation of glucose in tomato fruit. The content of glucose was decreased in the PMC-MEI anther and seems not increased even for the total amount of fructose and glucose.
Author’s response: Thank you for your observation. The authors considered your observation paramount and had incorporated a clear association between hexokinases with the contents of hexoses in the PMC-MEI anthers. See page 19-20 lines 690-698.
Below lines are added in revised manuscript:
Committing hexose sugars from sucrose cleavage to participate in starch biosynthesis and other metabolic processes requires their phosphorylation by hexokinase, demonstrated to be specifically expressed in the tapetum and pollen of developing anther (Granot et al., 2013; Suwabe et al., 2008). In this study, we observed two hexokinase genes, Slhxk1 and Slhxk2 significantly repressed in PMC-MEI anthers (Figure 6A) resulting in increased level of the monosaccharide sugar, fructose in agreement with (Hu et al., 2019). However, glucose was decreased in the PMC-MEI anthers suggesting its continued basal level utilization as energy source to fuel life-supporting biological processes during the DS, demonstrated by the resistance to drought stress and normal development of the PMC anthers after rewatering (Figure S2D).
4 -Figure 2D, E.
These data seemed redundant, since the rate of unset fruit might have the same meaning to the rate of flower abortion. How were these parameters, particularly fruit set, measured?
Author’s response: The results in Figures 2D and E are inversely relationship and serve as proof of the validity of the experimental data. They were generated so as to enable us validate whether the male was actually responsible for the poor fruit set/yield of the DS plants since the overall pollen viability for the DS plant was significantly lower compared to the well-watered plants (Figure 2A) and to determine whether DS affected the female organ significantly. The results in these figures confirmed that the male and not female was more susceptible to DS and was the main cause of the poor fruit yield performance of the DS plants and formed the basis for further cyto-histological, transcriptome and physiological investigations of the male organ in this study.
For determination of fruit set and flower abortion, the number of flowers involved in each pollination experiment was recorded. Of that total, the number of flowers that set fruit as well the number that aborted was estimated. Fruit set and flower abortion were estimated by expressing the total of flowers that set fruit and aborted respectively as percentages of the total number of flowers involved in each experiment. We already mentioned P3 L139-142
5 - Page 6 line 290-291.
I could not understand what did the following sentence, particularly the numbers, mean: "4" of them reaching anthesis during the DS period while "1" remained unopened... The authors should rephrase the sentence.
Author’s response: Thank you for your constructive suggestion. The authors made appropriate amendments so that the sentences on page 7 lines 292-298 can be understood, in accordance to your kind suggestion.
6 - Figure 5, 6, and 7.
The authors should clarify the calculation procedures for the "Relative expression levels" in the legends and/or the materials and methods, because it must be based on the RNA-Seq data.
Did they mean FPKM?
Author’s response: Thank you for your deep knowledge in the subject matter. A more thorough clarification procedures for the ‘Relative expression levels’ in the legends of Figures 5 ,6, and 7 have been made and they were based on FPKM values. See P14 L520-521; P16 L 550-551; P17 L 583-584.
7 - Page 13 line 480.
The authors referred here to "lipid-derived hormone metabolisms" repressed by drought, but it was not clear from the descriptions shown. The associated genes with hormones (JA?) in "lipid metabolic process" may support the suggestion.
Author’s response: Thank you very much. The authors followed your advice and indicated jasmonic acid as an example of a hormone derived from lipid metabolism. See page 13 line489-490.
Minor remarks
- Figure 1A The label "sampling time" indicated by an arrow is quite confusing, which may indicate sampling for RNA-Seq and not for data of this figure. The label should be removed or more properly explained.
Author’s response: Thank you very much for such a critical observation. The authors followed your advice and deleted sampling time in Figure 1A
- Figure 1E There is no scale bar.
Author’s response: Thank you very much. The authors noted your observation and added a scale bar to Figure 1E. It can be seen on P8 L333
- Figure 1E The authors should describe the timing of plant stage/age in the pictures.
Author’s response: Thank you for your suggestion. The authors followed your suggestion and described the floral buds in Figure 1E and stated the time of declaring them aborted. See page 8 line 341-343.
-Figure 2D The labels for crossing, male x male, need to be corrected.
Author’s response: The authors apologize for the mistakes you observed. We corrected them and placed the correct labels for crossing, female x male in Figure 2D.
- Figure 3A I would suggest to label the tapetum also in the EUM of WW.
Author’s response: The author followed your advice and labelled the tapetum layer in the EUM of WW section in Figure 3A
- Figure 8 The explanations for red arrows and dashed line are needed.
Author’s response: Thank you for your salient suggestions. The author followed your advice and provided explanations for the red arrows and dashed line. See P21 L748-751
- Page 3 line 123 This is the first place the abbreviation of drought stress, DS, appeared, thus it should be spelled out here. Please confirm the similar cases through the manuscript. For example, abbreviations of stage names, e.g. PMC, TED etc., and DARW were repeatedly spelled out and it reduced readability.
Author’s response: The authors are very grateful for your observation and advice. We followed your advice and drought stressed in full followed immediately by its abbreviation (DS) the first time it appeared in the manuscript. See page 3, line 123. We also cross-checked similar cases throughout the manuscript.
- Page 4 line 181 "RNA-Sequence" should be "RNA-Sequencing"
Author’s response: Thank you very much. The authors did change ‘RNA-Sequence’ to ‘RNA-Sequencing’ in accordance to your advice. See page 4, line 188.
- Page 4 line 185 "group I" should be "group 1"
Author’s response: Thank you very much very keen observation. In accordance to your advice, we changed ‘group l’ to ‘group 1’. See page 4, line 175.
- Page 6 line 254 typo "ddecrease"
Author’s response: Thank you very much. The authors heeded to your advice and corrected the typographical error from ‘ddecrease’ to ‘decrease’. See page 6, line 257.
- Page 6 line 257 For the preposition of DTRW and DARW, "on" sounds uncomfortable and "at" may be more suitable. Please confirm.
Author’s response: Thank you very much. The authors deep investigation in the internet and in a research article similar to ours (56), page 23 line 857, in respect of the preposition of DTRW and DARW, ‘on’ and confirmed that your suggestion was absolutely in place. We effected the changes from ‘on DTRW’ and ‘on DARW’ to at DTRW and at DARW’ not only on page 6, line 257 but in any other location in the manuscript.
- Page 6 line 266 "1 day after rewatering" should be "One day after rewatering"
Author’s response: Thank you very much for your kind suggestion. The author did change ‘1 day after rewatering’ to ‘one day after rewatering’ in accordance to your advice. See page 6 line 269.
- Page 7 line 316 typo "ADRW"
Author’s response: Thank you very much. The authors corrected the typographical error and changed ‘ADRW’ to ‘DARW’ . See page 7 line 322
- Page 9 line 349 The abbreviation of DAA should be days after anthesis.
Author’s response: Thank you very much for the observation. However, I am deeply sorry to inform you that the abbreviation DAA on page 9 line 356 actually refers to days to anthesis. Reason being that the floral buds labelled -1 to -4 DAA, oldest to youngest respectively in Figure S3A were in the bud state at -4 DTRW, opened from -3 to 0 DTRW, the first to the fourth day of the DS period respectively and had pollen viability very similar to that in WW plant in Figures 2B and C implying that the pollen in anthers of floral buds that are as old as -1 DAA to as young as -4 DAA are significantly not affected when drought stressed.
- Page 9 line 370 Brief explanation of pollen mitosis 1 here or in the materials and methods.
Author’s response: Thank you very much for the good suggestion. The authors provided a brief explanation of pollen mitosis 1 in accordance to your suggestion. See page 9 lines 378-379.
- Page 9 line 382-386 The citation positions for Figure 2D and E should be corrected.
Author’s response: Thank you very much for the excellent observation. The authors corrected the citation ‘Figure 2E’ to ‘Figure 2D and E. See page 9 line 393-395.
Reviewer 2 Report
In this manuscript, the authors have studied the impact of drought on floral development, specifically on pollen development, in tomato through phenotypic, physiological, and transcriptome analyses. The authors have done a good job with detailed morpho-physiological experiments and supported the outcomes of RNAseq analysis with various hormones and starch/sucrose measurements. I would recommend this manuscript for publication.
Suggestions:
I encourage authors to upload the raw RNA sequencing reads/data to the public database so that readers can reproduce the transcriptome results.
Author Response
Reviewer 2
I encourage authors to upload the raw RNA sequencing reads/data to the public database so that readers can reproduce the transcriptome results.
Author’s response: The authors appreciate your suggestion; the RNA sequence data have been uploaded to NCBI. When we get the accession number we will add in the proofreading of this manuscript.
Below sentence is added in the revised manuscript
The raw sequence reads were deposited into NCBI Sequence Read Archive under accession the number XXXX (the number will be added after the allotment in the final version of the manuscript). It can be seen P5 L200-202